# Tailored Triggering of High-Quality Multi-Dimensional Coupled Topological States in Valley Photonic Crystals

**DOI:** 10.3390/nano14100885

**Published:** 2024-05-19

**Authors:** Guangxu Su, Jiangle He, Xiaofei Ye, Hengming Yao, Yaxuan Li, Junzheng Hu, Minghui Lu, Peng Zhan, Fanxin Liu

**Affiliations:** 1Department of Applied Physics, Zhejiang University of Technology, Hangzhou 310023, China; gxsu@zjut.edu.cn (G.S.); 2112109030@zjut.edu.cn (J.H.); 211123090024@zjut.edu.cn (H.Y.); 211122090031@zjut.edu.cn (Y.L.); 2National Laboratory of Solid State Microstructures, Collaborative Innovation Center of Advanced Microstructures, School of Physics, Nanjing University, Nanjing 210093, China; mf21220006@smail.nju.edu.cn (X.Y.); dg21220019@smail.nju.edu.cn (J.H.); 3National Laboratory of Solid State Microstructures, Department of Materials Science and Engineering, Nanjing University, Nanjing 210093, China; luminghui@nju.edu.cn

**Keywords:** optical microcavity, multi-dimensional coupled topological states, higher-order photonic topological insulators, valley photonic crystals, pseudospin dependence

## Abstract

The combination of higher-order topological insulators and valley photonic crystals has recently aroused extensive attentions due to the great potential in flexible and efficient optical field manipulations. Here, we computationally propose a photonic device for the 1550 nm communication band, in which the topologically protected electromagnetic modes with high quality can be selectively triggered and modulated on demand. Through introducing two valley photonic crystal units without any structural alteration, we successfully achieve multi-dimensional coupled topological states thanks to the diverse electromagnetic characteristics of two valley edge states. According to the simulations, the constructed topological photonic devices can realize Fano lines on the spectrum and show high-quality localized modes by tuning the coupling strength between the zero-dimensional valley corner states and the one-dimensional valley edge states. Furthermore, we extend the valley-locked properties of edge states to higher-order valley topological insulators, where the selected corner states can be directionally excited by chiral source. More interestingly, we find that the modulation of multi-dimensional coupled photonic topological states with pseudospin dependence become more efficient compared with those uncoupled modes. This work presents a valuable approach for multi-dimensional optical field manipulation, which may support potential applications in on-chip integrated nanophotonic devices.

## 1. Introduction

The discovery of Chern insulators and a series of proposed topological effects in condensed matter physics has driven the development of topological photonics [1,2,3], which brings new avenues for transmitting and localizing light [4,5]. Photonic crystals are analogs of conventional crystals that replace the atomic lattice with a periodic medium, providing an excellent platform for topological physics due to the controllability band structure [6]. In practice, defects and impurities are inevitably introduced in the sample preparation, leading to energy loss and signal distortion. In the face of the above difficulties, the topologically protected photonic states are proposed and demonstrated to be of benefit for the dissipationless transport dynamics of light [7,8,9,10]. More recently, photonic higher-order topological insulators (HOTIs) with bulk-edge-corner correspondences have been extensively studied for their ability to control light in multi-dimensions, in which the topological index can be characteristic by the vectored Zak phase and Wannier center [11,12,13,14,15,16]. In addition, by introducing coupling effects between a series of topological states, the quality of a nanocavity can be further improved, which also brings extra freedom to manipulate light [17,18,19,20]. However, the photonic topological edge states and lower-dimensional corner states in HOTIs tend to be discrete in spectrum. To realize multi-dimensional coupling, the structure needs to introduce unit distortion or more complex artificial design, which would narrow the bandgap of bulk and limit the development of related applications.

Valley photonic crystals (VPCs) with non-zero Berry curvature in momentum space [21,22,23] provide a new method to realize higher-order topological phases, which has already been successfully demonstrated in many lattice structures such as kagome lattices [24], triangular lattices [25], honeycomb lattices [26], and square lattices [27]. Among them, the supported valley corner states (VCSs) are robust and valley-locked-dependent [28,29,30]. Based on this feature, several interesting optical devices have been designed, such as topological all-optical switches [31] and topological rainbows [32]. By combining HOTIs and valley freedom, the structure can both support diverse types of topological states, which may support potential applications in topological lasers, topological optical switches, and on-chip integrated optical circuits.

In this work, we computationally propose an on-chip photonic device for the 1550 nm communication band, in which the multi-dimensional coupled topological states are achieved with two types of VPCs unit. By optimizing the coupling strength between the zero-dimensional VCSs and the one-dimensional valley edge states (VESs), the transmission spectrum presents a typical Fano line, showing as a high-quality localized mode. Furthermore, we extend the valley Hall effect of light to a high-order version, and successfully visualize the directed excitations of coupled VCSs with different chiral sources. The simulated results provide a versatile way to manipulate light based on VCSs and VESs, which can also extend to other electromagnetic wave ranges by adjusting the structure size. By the way, the valuable approach for multi-dimensional optical field manipulation can extend to other material systems, such as GaAs or InP, as long as we replace the corresponding refractive index parameters and fine-tune the structure parameters.

## 2. Results and Discussion

The designed VPC sample is a silicon on insulator (SOI) with a 220 nm thick silicon layer and specific periodic holes. As shown in Figure 1a, the valley photonic structure is arranged in a honeycomb lattice with a=470 nm period, and there are two rounded equilateral triangular air holes with side lengths of $l_{1}$ and $l_{2}$ in a single cell, where $\delta =l_{1}-l_{2}$ and $l_{1}+l_{2}=a$. For triangular holes, VPC has a larger band gap compared to the circular holes. And the effective refractive index of silicon is defined as 2.83. In all numerical calculations, we use the commercial software COMSOL Multiphysics 5.4 based on the finite element method. The periodic boundary conditions and scattering boundary conditions are used for corresponding interfaces. And the mesh is set up as the build-in physical field segmentations. The calculated band structures of VPCs under transverse electric (TE) polarization are as shown in Figure 1. When δ=0, the VPC has $C_{6v}$ symmetry and the corresponding band structure presents an obvious degenerate Dirac point at the K($K^{\prime }$) valley. When δ≠0, the VPC changes to $C_{3}$ symmetry and the Dirac cone can be gapped out, leading to two valley states with opposite circularly polarized chirality at the two unequal K($K^{\prime }$) valleys. For $l_{1}>l_{2}$, the K valley state in the first band is a right-handed circularly polarized (RCP) mode and the K valley state in the second band is a left-handed circularly polarized (LCP) mode. For $l_{1}<l_{2}$, the two chiral polarizations of the K valley state are inverted, indicating the topological phase transition. Here, we designed two VPC units with δ=0.6a (${VPC}_{1}$) and δ=−0.6a (${VPC}_{2}$), which have the same band structure but different topological phases. Furthermore, we numerically calculated the Berry curvature of the first band:$$\Omega_{n}\left(k\right)=\nabla_{k}\times A_{n}\left(k\right)=\frac{\partial A_{y}\left(k\right)}{\partial k_{x}}-\frac{\partial A_{x}\left(k\right)}{\partial k_{y}}$$
where $A_{n}\left(k\right)=i\left\langle u_{n,k}|\nabla_{k}|u_{n,k}\right\rangle$ is the Berry connection and $\left|\left.u_{n,k}\right\rangle \right.$ is the Bloch periodic function. Although the VPCs do not break the time-reversal symmetry, the Chern number of the band is zero and the valley Chern number at the K($K^{\prime }$) valley is nonzero due to the breaking of the space-reversal symmetry. As shown in Figure 1d, the Berry curvature of ${VPC}_{1}$ near the K($K^{\prime }$) valley is greater (less) than zero, while the Berry curvature of ${VPC}_{2}$ near the K($K^{\prime }$) valley is less (greater) than zero. For the same band, the ${VPC}_{1}$ and ${VPC}_{2}$ satisfy Ω(k)=−Ω(−k). The integral of the Berry curvature is calculated as shown below:$$C_{K/K^{\prime }}=\frac{1}{2\pi }\int_{{HBZ}_{K/K^{\prime }}}\Omega_{n}\left(k\right)d^{2}k$$

The valley Chern number of our system is half-integer, $C_{K}=-C_{K^{\prime }}=1/2$ for ${VPC}_{1}$ and $C_{K}=-C_{K^{\prime }}=-1/2$ for ${VPC}_{2}$, indicating two opposite topological phases.

Next, we demonstrate two topological VESs based on the ${VPC}_{1}$ and ${VPC}_{2}$. For type I splicing interface, as shown in the left of Figure 2c, the larger triangular holes at the splicing interface are edge to edge. Meanwhile, for type II splicing interface, as shown in the right of Figure 2c, the larger triangular holes at the splicing interface are cusp to cusp. The projected band structures of two types of splicing structure are as shown in Figure 2; there is a bandgap between the type II VES (dashed line) and upper bulk states, which provides a possible coupling effect between VCS (orange dashed line) and another type I VES (solid line). In this case, the structure does not need to introduce any unit distortion or more complex artificial design. In addition, the valley-dependent VESs have opposite group velocities and vortices near the K valley and $K^{\prime }$ valley, which originates from the valley pseudospin-momentum locking. In order to visualize this physical feature, we calculated the two edge states at the K valley (labeled with different colored diamond symbols in Figure 2a) with the Poynting vectors and $H_{z}$ phases as in Figure 2c. For the upper and lower interfaces of the splicing interface, the energy flow direction and phase vortex direction at K valley are opposite. And the dependencies for two types of VESs are also opposite, indicating the valley-momentum locking properties. Due to the time-reversal symmetry, the $K^{\prime }$ valley of the same structure has similar dependencies, as shown in Appendix A.

Based on the feature, we can achieve a directional transmission of light by using different circular polarizations of light. Here, we define the Stocks parameter to quantitatively analyze the unidirectional transmission ability of VESs as shown below [33,34]: $$D=\frac{P_{R}-P_{L}}{P_{R}+P_{L}}=\frac{S_{3}}{S_{0}}=\frac{2Im\left({E_{x}^{*}E}_{y}\right)}{{\left|E_{x}\right|}^{2}+{\left|E_{y}\right|}^{2}}$$

Taking the type I VES as an example, $S_{0}={\left|E_{x}\right|}^{2}+{\left|E_{y}\right|}^{2}$ and $S_{3}=2Im(E_{x}^{*}E_{y})$ indicates the circular polarization point of the local polarization as the RCP (LCP), respectively. $P_{R}$ ($P_{L}$) represent the energy of light transmitting to the right (left) and D=±1 represents that, ideally, light is transmitted completely to the right or left. 

As shown in Figure 2b,d, we place an RCP source in three typical areas near the splicing interface; position 1 and 3 are the upper zone with D=1 and the bottom zone with D=−1, respectively. Position 2 is the center of the interface with D=0, which means that the unidirectional transmission of energy becomes worse. In simulation, we detected the energy of light at left or right ports to verify the directional transmission capability and the simulated field distributions agree well with the theoretical predictions. When an RCP source is placed at position 1, the energy of electromagnetic waves from frequency 164 to 198 THz will transmit to the right, where the light leaves through the right output port. And the energy will transmit to the left when the RCP source is placed at position 3 or is replaced by LCP source. When the RCP source is put to position 2, the transmission of light to both sides is almost equal. 

The valley-momentum locking mentioned above can be understood from the quantized valley Chern number of the VESs. For the type I VES, the valley Chern number can be defined as $C_{IK}=C_{{VPC}_{2}K}-C_{{VPC}_{1}K}=-0.5-0.5=-1$,$C_{{IK}^{\prime }}=C_{{VPC}_{2}K^{\prime }}-C_{{VPC}_{1}K^{\prime }}=1$. Similarly, the valley Chern number of the type II edge state can be defined as $C_{IIK}=C_{{VPC}_{1}K}-C_{{VPC}_{2}K}=0.5-(-0.5)=1$,$C_{IIK^{\prime }}=C_{{VPC}_{1}K^{\prime }}-C_{{VPC}_{2}K^{\prime }}=-1$. Valley Chren numbers with the same sign have consistent valley dependence properties and vice versa. More importantly, the quantizable valley-momentum locking properties of VESs and the naturalness of coupling with VCSs are the keys to the next realization of nanocavities with high responsiveness and high performance.

By combining the HOTI and valley degrees of freedom, the splicing corners of two VPCs with different valley Chern numbers can excite the VCS modes due to the valley–valley interactions of the VESs. Here, we construct a trapezoidal splicing structure, where the ${VPC}_{1}$ unit is surrounded by ${VPC}_{2}$ unit. As shown in Figure 3a–c, although there are four splicing corners here, only two VCS can be support in the eigenmodes simulations. And the 60-degree splicing corner appears obvious, while 120-degree angle disappears. The selective activation of VCS is related to the sign flip of the valley Chern number at the splicing corners and more details can be found in our previous work [31]. Due to the collective coupling effects of two VCSs, the equivalent corners of the structure appear to be two asymmetric VCSs, leading to spectrum division. The bonding coupled VCS with lower frequency presents two synchronous nanocavities, named as $\varphi_{C}^{+}=\varphi_{1}+\varphi_{1}^{\prime }$. Meanwhile, for the anti-bonding coupled VCS with higher frequency, the adjacent nanocavities present a π phase difference, named as $\varphi_{C}^{-}=\varphi_{1}-\varphi_{1}^{\prime }$. The physics behind this can be referred to the electrodynamics theory; when two electric dipoles in the same direction end to end are close to each other, the two dipoles attract each other to form a bond and the energy of the coupled system decreases, corresponding to the bonding coupled modes with lower eigenfrequency. Similarly, when two dipoles in the opposite direction end to end are close to each other, the two dipoles are mutually exclusive to form an anti-bond. Furthermore, we have extended the valley Hall effect of light to a high-order version, as shown in Figure 3d–f. When an RCP source is placed on the center of the splicing interface, the intensity at point B is slightly stronger than point A. The directional excitation of the nanocavity will be reversed once the LCP source is placed on the same position. This interesting phenomenon is related to the valley-momentum locking and pseudospin polarization of the two coupled VCSs. It is worth noting that, although there are two eigenvalues of the VCS in our system, only one resonant peak is observed, which might be attributed to the small difference between the two eigenfrequencies. Next, we have calculated the quality factor Q of the nanocavity, which can be expressed as $Q=\frac{\omega_{0}\tau }{2}=\frac{\omega_{0}}{∆\omega }$ in terms of the resonance frequency ($\omega_{0}$) and the decay time of the electromagnetic energy in the cavity (τ) or the resonance linewidth (∆ω) and the normalized field-strength spectrum, which shows that the quality factor of the nanocavities are around 375. On the other hand, the energy conversion from source to the VCSs is also weak.

To further enhance the quality factor and response of VCSs, we have designed a waveguide-nanocavity coupled structure based on the ${VPC}_{1}$ and ${VPC}_{2}$, as shown in the inset of Figure 4a, where the zero-dimensional corner state and one-dimensional edge state can be naturally coupled. Here, the VES with an odd or even symmetric field distribution are defined as $\varphi_{E}^{+}\;or\;\varphi_{E}^{-}$. Since the two wave functions with different symmetries are orthogonal to each other, $\left\langle \varphi_{C}^{+\left(-\right)}|\varphi_{E}^{-\left(+\right)}\right\rangle =0$, there are only four multi-dimensional coupled modes in our system, as shown in Figure 4. For ${Mode}_{1}=\varphi_{C}^{+}+\varphi_{E}^{+}$, the eigenfrequency is obviously lowest, corresponding to the bonding coupling between VES and VCS. And the eigenfrequency of ${Mode}_{4}=\varphi_{C}^{-}-\varphi_{E}^{-}$ is highest, corresponding to the anti-bonding coupling between VES and VCS. As for the multi-dimensional coupled topological states with frequencies in between, the modes are defined as ${Mode}_{2}=\varphi_{C}^{-}+\varphi_{E}^{-}$ and ${Mode}_{3}={-\varphi }_{C}^{+}+\varphi_{E}^{+}$. These multi-dimensional coupled topological states can be distinguished from the field distributions, where the energy of ${Mode}_{1}$ and ${Mode}_{4}$ are mainly concentrated in the VESs, shown as bright modes, while the energy of ${Mode}_{2}$ and ${Mode}_{3}$ are mainly concentrated in the VCSs, shown as dark modes. By introducing coupling effects between the VCS and VES, the quality of nanocavities can be further improved, which also brings extra freedom to manipulate light on chip.

As shown in Figure 5, the quality factor and directional transmission capability of coupled VCSs have been significantly improved. When an RCP source is placed at position 1, the field intensity at the right corner is much stronger than the left corner in a wide wave range. If an LCP source is used, only the left and right corners are switched due to the time-reversal symmetry. When we only change the spatial position of the RCP source on the splice interfaces, the other structures are unchanged. And the results show that the field intensities of the right and left corner states are almost the same no matter which chiral source is placed at position 2. When RCP source is placed at position 3, the intensity of the left corner is much stronger than the right corners in a wide wave range. In the meantime, the intensity at corresponding corners is increased by 2–3 orders of magnitude compared to uncoupled VCS, as visualized in Figure 3 and Figure 5. The reason for this high responsiveness and higher-order topological valley-locked characteristic is because the energy of the VCS at this point is directly affected by the selective coupling of the topological waveguide, and the responsiveness of the chiral source to the selective excitation of the VCS is equivalent to perturbation, i.e., the unidirectional transmission capability of the VES determines the field strength ratio at the two splice corners on both sides. 

It is worth noting that, when the frequency of the excitation source is near the resonance frequency of the nanocavity, the field strengths on both sides of the splice corners are almost the same, which is due to the fact that the introduction of the nanocavity inevitably disrupts the overall symmetry of the lattice, and the unidirectional transmission ability of the VES is limited at the resonance frequency of the VCS, as shown in Figure 5e,g, where the topological waveguide still maintains a good unidirectional transmission capability, while the magnitude of the energy flow density on both sides is almost the same near the resonance frequency. In order to further characterize the excellent performance of the valley nanocavities, we calculate their quality factors, and the results show that both sides of the nanocavities have very high-quality factors in all three cases up to about 20,000, which is nearly 50 times higher than that of the previous uncoupled system. 

For the well-designed topologically protected nanophotonic devices with high responsiveness and high performance, the system also shows a tunable asymmetric spectral line in the transmission spectrum. And the physical mechanism behind it is the Fano resonance phenomenon arising from coherent interference between the discrete coupled VCS and the continuous VES near the resonance frequency, which requires that the resonance frequency of the discrete state is in the frequency range of the continuous state. Unlike other schemes, our structure naturally satisfies this condition without changing any parameters. As shown in Figure 5e–g, when we place the RCP source in positions 1 or 3, the transmission spectra at the left port present a typical Fano line shape, while the transmission spectra at the right port shows an electromagnetically induced transparency-like (EIT-like) line shape. If the source is placed at position 2, the transmission spectra of the ports on both sides are EIT-like line shapes. By considering the field distribution at different wavelengths, we can intuitively understand the above spectral response as shown in Appendix A. This reveals that the Fano and EIT-like resonance phenomenon originates from the coherent interference between the edge–corner coupling states with different line widths, where the modes on the wide transmission spectral lines are all ${Mode}_{1}$ or ${Mode}_{4}$ with energy concentrated in the topological waveguide, corresponding to the bright modes, and the modes on the narrow spectral lines are all ${Mode}_{2}$ or ${Mode}_{3}$ with energy concentrated in the valley nanocavity, corresponding to the dark modes. The Fano resonance is formed by the destructive and constructive interference of the two modes. On the other hand, since the system maintains the time-reversal symmetry, when we place the LCP source at different positions of the splicing line, the result is only that the spectral lines are swapped, while the other laws remain the same, as shown in Appendix A. In addition to the advantage of tunability, our system is also extremely robust. As shown in Appendix A, where we destroy the geometry near the nanocavity by replacing it from the original small nanopore to a large one, the calculation results show that only the resonance frequency of the cavity is slightly blue shifted, while the cavity’s localization, quality factor, and higher-order valley-locking properties are basically unaffected, and the Fano resonance spectral lines are also relatively well protected. In conclusion, this system and modulation we have established not only greatly improves the quality factor of the valley nanocavity and the responsiveness of the higher-order valley-locking properties but also achieves topologically protected tunable Fano and EIT-like resonance spectra in the same structure.

Finally, we also demonstrate the modulation of the system by changing the coupling distance between the cavity and the waveguide. Figure 6a,c shows the evolution of the quality factors of the left and right cavities with the coupling distance when the right-handed chiral source is in the splicing line at positions 1 and 2, and Figure 6b,d show the transmission spectra corresponding to the left and right ends. The relevant data when the RCP source is at position 3 are shown in Appendix A. These results show that, as the coupling strength decreases, the quality factor of the cavities increases roughly linearly, up to about 60,000. From the transmission spectra, we can derive that the resonance frequencies of the dark modes are gradually blue-shifted, and the peaks of the Fano spectral line and the EIT-like spectral line are also reduced substantially; however, their shapes are basically maintained in the same way. In addition to the coupling distance of four cell sizes, the transmission spectra of the RCP source are all classical Fano lines when placed at positions 1 and 3 and the transmission spectra are all EIT-like lines when placed at position 2, which provides a new method and path for the modulation of the topological Fano transmission spectral lines.

## 3. Conclusions

In summary, we computationally propose a topologically protected high-quality optical nanocavity, which can be selectively triggered and modulated on demand. Based on the mismatch in the spectrum of two valley edge states, we successfully demonstrate the coupling effect between the zero-dimensional valley corner states and the one-dimensional valley edge states without any structural alteration. By optimizing the coupling strength between the valley corner states and edge states, we observe an extremely high-quality localized mode. Furthermore, we have extended the valley Hall effect of light to a higher-order version, where the selected photonic topological corner states can be directionally excited with different polarizations of light and the coupled VCS with pseudospin dependence become more efficient. This work visualizes an efficient and flexible electromagnetic mode with pseudospin dependence, which is valuable for the development of on-chip integrated topological photonic devices.

## Figures and Tables

**Figure 1 nanomaterials-14-00885-f001:**
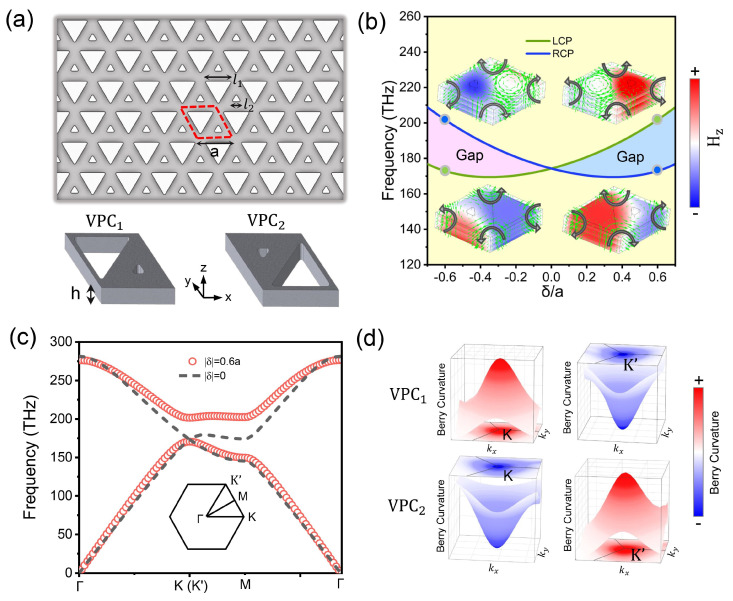
Topological phase transition and band structure. (**a**) Schematic of the VPC; the red dashed lines are the initial unit cell of the VPC. (**b**) A phase diagram showing the variation of the band gap as a function of δ. The inset shows the field distribution and Poynting vectors at selected points, shown with green and blue points; the rose and blue area corresponds to two opposite topological phases. (**c**) The band structures for VPCs. (**d**) Distribution of Berry curvature around K valley and $K^{\prime }$ valley for ${VPC}_{1}$ and ${VPC}_{2}$.

**Figure 2 nanomaterials-14-00885-f002:**
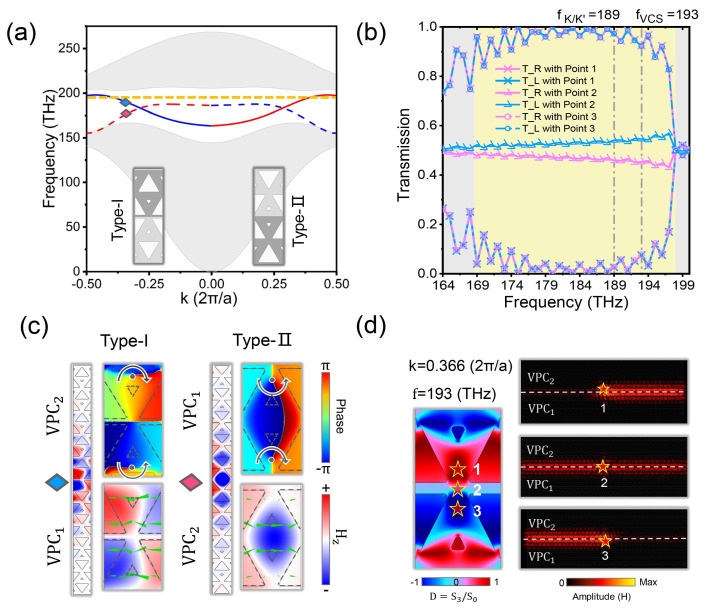
Topological projection band diagrams of VESs and valley-momentum locking phenomenon. (**a**) Band structure for the type I and type Ⅱ interfaces; the solid and dashed line are the eigenmode of type I and II VES, respectively. And the grey area is bulk modes. (**b**) Normalized energy flow at the left or right ports when an RCP source is at positions 1, 2, and 3. (**c**) Valley-momentum locking properties of two types of VESs at K valley, which is selected from (**a**). (**d**) Positional dependence of the normalized Stokes $S_{3}/S_{0}$ parameter and the field distributions with the RCP source at positions 1, 2, and 3.

**Figure 3 nanomaterials-14-00885-f003:**
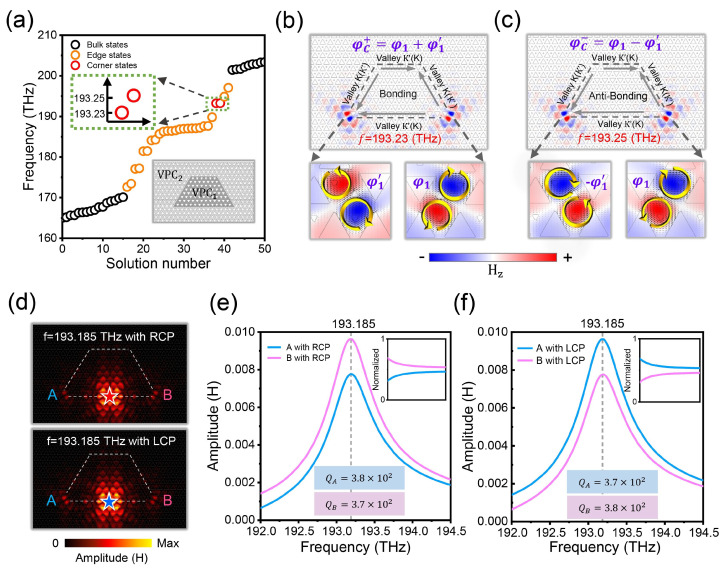
Pseudospin dependence of VCSs. (**a**) Eigenvalues of the bulk, edge, and corner states. (**b**,**c**) Field distribution of the coupled VCSs; arrows represent the direction of current. (**d**) Field distributions with different chiral sources at resonance frequencies, red (blue) stars are right-handed (left-handed) sources, respectively. (**e**,**f**) Field intensity at points A and B with an RCP or LCP source at the center position.

**Figure 4 nanomaterials-14-00885-f004:**
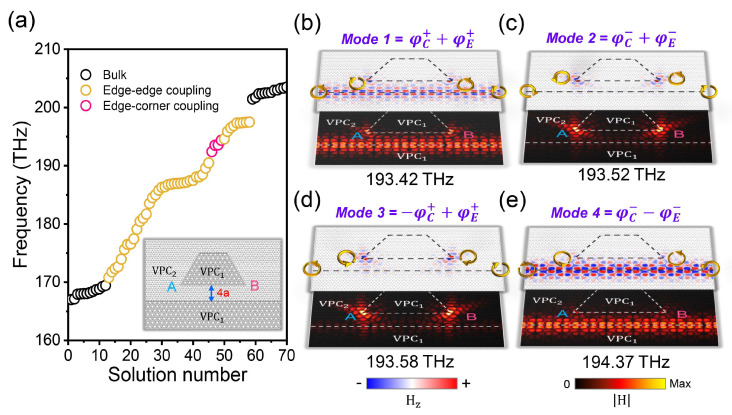
Characteristic of multi-dimensional coupled topological states. (**a**) Eigenvalues of the bulk, edge, and corner states; the inset shows a schematic structure. (**b**–**e**) Field distributions of four coupled topological states.

**Figure 5 nanomaterials-14-00885-f005:**
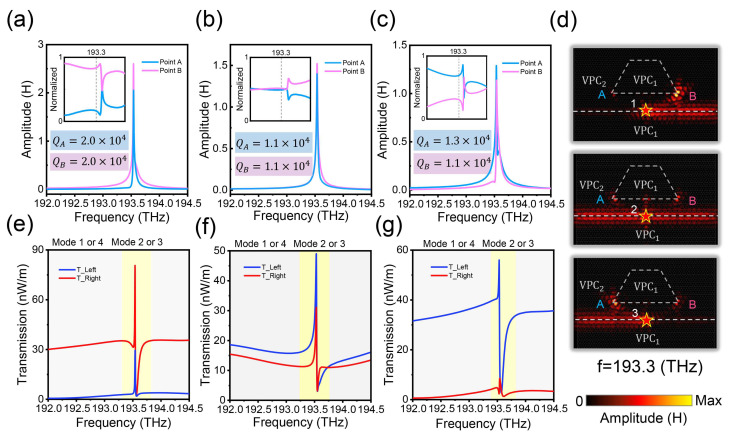
Pseudospin dependence of high-quality multi-dimensional coupled topological states. (**a**–**c**) Field intensities of the VCS nanocavities at points A and B when an RCP source is at positions 1, 2, and 3, respectively, and the inset shows the normalized field intensities. (**d**) Field distributions of the RCP source at the 1, 2, and 3 positions. (**e**–**g**) Transmission spectrum at the left or right ports when an RCP source is at positions 1, 2, and 3.

**Figure 6 nanomaterials-14-00885-f006:**
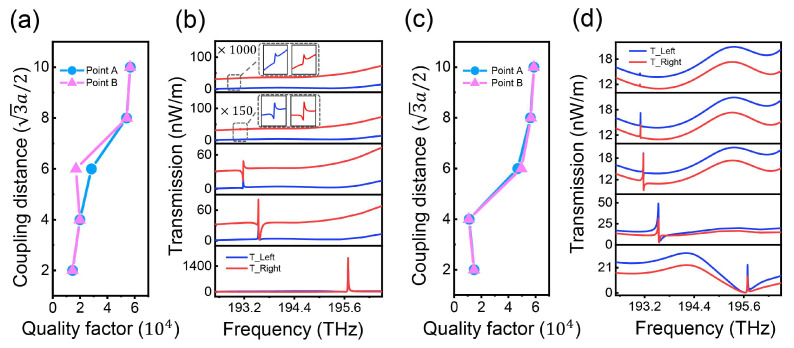
Regulation of multi-dimensional coupled topological states with different coupling strength. (**a**,**b**) Dependence of the quality factor and transmission spectrum of the structure with different coupling strength when the RCP source is at position 1. (**c**,**d**) Corresponding parameters when the RCP source is at position 2.

## Data Availability

The data that support the findings of this study are available from the corresponding author upon reasonable request.

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
