# Peer review of "Tailored Triggering of High-Quality Multi-Dimensional Coupled Topological States in Valley Photonic Crystals"

_nanomaterials, 2024, doi:10.3390/nano14100885_

Round 1
Reviewer 1 Report
Comments and Suggestions for Authors
The manuscript goes deeply into theoretical explanation of topological states in valley photonic crystals. There is quite a lot work visible behind the results, but I have some concerns to the presentation of the results. Some decisions of the authors are not clear to me. For example the reason to choose the triangle shapes for the valley photonic crystal, it shall be mentioned in the text. I also lack the motivation behind the analysis of such structures. The only one mention of wavelength 1550nm of communication band and the frequency ranges in graphs do not explain the motivation because the work is purely theoretical and the choice of wavelength does change only the dimensions of the structure, but not the results. Why are triangles better than for example circles and for what purpose? Do you realize, that such structure will be very challenging for fabrication and thus almost impossible to apply the study in practice? Furthermore, what was the reason for choosing the design of the waveguide nanocavity coupled structure? The short explanation "To further enhance the quality factor and response ..." seems to me insufficient for such complex design, I would expect more deep analysis. To summarize the general complains, I am also not really convinced that the manuscript has been submitted to correct journal. The work is purely theoretical with strong mathematical explanations and for me it is a high quality analysis for publication in math oriented journals.
There are some specific things in the content which should be corrected. In row 149 the sentence begins with "Table 3. a-c) .... " which is not in context and should be corrected. When I look at the detail in Fig. 1a) I cannot believe that "a" is equal to "l1+l2" as mentioned in row 76. In Fig. 2 a) is missing the legend for type I and II VES and VCS. The figures are generally very complex and the quality of the reviewed document was very low to see the details which caused high confusion for me. Please, assure that in the final version of the paper the quality of the pictures will be higher or redraw the pictures to better highlight the thin lines which are important for reader's comprehension of the context.
If the authors will correct these concerns, I will recommend the manuscript to be published.
Reviewer 2 Report
Comments and Suggestions for Authors
Ref.comments to the paper titled as “ Tailored Triggering of High-quality Multi-dimensional Coupled Topological States in Valley Photonic Crystals” written by the authors: Guangxu Su, Jiangle He, Xiaofei Ye, Hengming Yao, YaxuanLi, Junzheng Hu, Minghui Lu,Peng Zhan and Fanxin Liu.
It is well known that the study of the photonic crystals due to their unique features for the realization of the different tasks is very important and modern. From this point of view, the current paper is actual.
For the first, this paper is included the analysis of 34 literature data. Indeed, the authors are known the problem and can make some steps to resolve the important tasks. Moreover, the analysis of the publications observed on last 3-5 years has been made carefully as well.
The paper is good illustrated; the pictures shown can help to the reader to understand the article text with good advantage.
Some remarks.
The remark about the understanding.
1). It is well known that the photonic crystals are the periodic structures with a tailored refractive index, they exhibit classical and novel optical characteristics that can be exploited for manipulating light at the nanoscale. In this concern, the optical processes are indeed semiclassically can be described by Maxwell’s equations, which include the nonlinear terms raising from nonlinear dielectric polarization. How does your consideration of the process allow you to add any change to the integral value of the polarization of the system? How will the cubic nonlinearity change, for example, in your case?
2). Please add in your consideration the real crystal, for example: InGaAs/GaAs, InGaAsP/InP, in which the Valley Hall effect and other manipulations with light can also be observed.
3). At which temperature regime your effects and your manipulations with light can be revealed?
Little simple remarks.
4). Section 2. Results and discussion. Please use Latin symbol (lattice parameter, others,…) as tilted one. Only Greek symbol can be straight.
5). Conclusion part is so little. Please extend this section.
As for my local opinion, the paper can be published after mijor corrections.

Reviewer 3 Report
Comments and Suggestions for Authors
Dear authors,
Thank you for the interesting work. Your findings considering interaction of circularly polarized light with topological interface inside photonic crystal make sense, and they worths sharing with scientific community through publication in Nanomaterials. I hope you agree with the statement.
Unfortunately, the paper couldn’t be published in the present state, because of the gross omissions resulting in text that can hardly be understood. Would you be so kind to consider:
1. Certainly, the computational design is important step in the development of any new technique. With simulations, one can model the performance of new functionalities, and foresee the potential problems as well. In this light, both title and abstract of the paper (and bigger part of introduction) entirely skip the computational character of the work, that will decrease the visibility of the paper for the relevant readers. At first sight the paper could be confused for the development and testing of the real device, that could be also disappointing for the readers really interested in this kind of the development.
I suggest reviewing the title and the abstract aiming to outline clearly that entire work is dedicated to the computational simulations of the new device with the new functions.
2. The description of the computations is missing entirely. Usually, the chapter dedicated to the setup of computations describes the computed characteristics against the specific software used for their calculations with the relevant details. For example, “The components of magnetic field in the electromagnetic wave were computed using the Comsol Multiphysics with the Wave Optics module. The mesh was setup as follows: …”. These details are important to disclose your results for the scientific community, since they prove these results are reproducible.
In case the custom written software or scripts were used, for example - to calculate the characteristics not available in the proprietary software like Comsol, this should be outlined as well.
3. On the page 5, line 149 starts with the reference to the Table 3, but I can’t find any table neither in the paper nor in the supplementary material. The text that follows is not connected well with the previous paragraph, so probably – there are 3 tables messing, and a piece of text as well.
This is for the major revision of the paper, and in any case, I can’t advance beyond the point where the content is missing. Would you be so kind to review?
In addition, I can suggest the revisions of figures 1 and 2, as well as their referencing in the text.
Figure 1
1. On the figure 1a, the lattice should be composed of the triangular holes with different sizes, because this represents the subject of the paper. In other words, the current figure 1a is not about the paper subject. This will require the appropriate correction of the dashed lines.
2. Please explain the meaning of the dashed lines on the figure 1a
3. On the figure 1b, the arrows showing the value and direction of Poynting vector are not visible, probably, they could be removed.
4. Figure 1b, in the description, the field is shown inside the whole elementary cell of the lattice, and not only in the selected points.
5. On figure 1b, what is the yellow area? What is the rose area? What is the blue area? What do the “Gap” titles mean?
6. On figure 1c, the legend has the font that is too small.
7. On figure 1d, the labels marking the K and K’ point are too small and difficult to see
Figure 2
1. The figure 2 should have first a scheme of two types of VES interfaces, type 1 and type 2.
2. Figure 2a, the description (legend) is missing. What is the yellow dashed curve and two blue-red curves dashed and continuous. The same applies to the grey areas. The text is not more clear on the subject.
3. Figure 2b, the font of legend for curves is excessively small
4. The figure 2b has elements not described, either in the figure caption or in text. What is the grey area and the yellow area? What is the “corner” frequency, fcorner?
5. What are the “ports” on the lattice structure, right and left? No figure or text explains the input or output port, these ports appear in the caption of the figure 2b. Is it simpler to mention that the light is directed to the right, or to the left?
6. On figure 2c placing on top the VPC2 or VPC1 complicates understanding by adding more confusion. I would strongly suggest keeping to “type 1” and “type 2” VES interfaces instead of “VPC1 on top” or “VPC2 on top”, or alike. In this light the figure 2c would be less confusing with the same bottom-up positioning of VPC1 and VPC2 for different VES interfaces.
7. Instead, on the figure 2c the geometry of the type-1 and type-2 VES interfaces should be outlined better. In the current version the lines depicting the triangular lattice underneath the magnetic field/phase distribution are very pale and are hardly visible.
8. Again, the arrows showing the value and the direction of Poynting vector on the figure 1c are hardly visible.
9. What is polarisation of light on the figure 2c?
I wish you to have a good continuation of the work.
Round 2
Reviewer 2 Report
Comments and Suggestions for Authors
Ref.comments to the paper titled as “Tailored Triggering of High-quality Multi-dimensional Coupled Topological States in Valley Photonic Crystals” written by the authors: Guangxu Su, Jiangle He, Xiaofei Ye, Hengming Yao, Yaxuan Li, Junzheng Hu, Minghui Lu, Peng Zhan and Fanxin Liu
I would like to repeat my previously made remarks about the actuality of the current paper:
It is well known that the study of the photonic crystals due to their unique features for the realization of the different tasks is very important and modern. From this point of view, the current paper is actual.
For the first, this paper is included the analysis of 34 literature data. Indeed, the authors are known the problem and can make some steps to resolve the important tasks. Moreover, the analysis of the publications observed on last 3-5 years has been made carefully as well.
The paper is good illustrated; the pictures shown can help to the reader to understand the article text with good advantage.
About author’s answers. Good. I have seen the revised version of the paper. The Results and Discussion part are modified with good advantage; the Conclusion sections is extended a little bit. I am agreeing!
Thus, as for my local opinion, the current paper can be published in the current form.

Author Response
We are grateful to the reviewer for her/his positive assessment of our work and recommending the manuscript for publication in Nanomaterials.
Reviewer 3 Report
Comments and Suggestions for Authors
Dear authors,
Thank you for considering the improvements, the paper is
Unfortunately, the paper still needs further revision:
1. The important point is presentation of the proposed photonic structure, in the current state it is difficult to understand. The proposed structure should be better presented on the figure 3. The text should be clarified, see the remarks in the attached file.
2. Please summarize in the conclusion the benefits of the proposed photonic structure with respect to the background.
3. I would strongly suggest keeping to “type 1” and “type 2” VES interfaces instead of “VPC1 on top” or “VPC2 on top”, or alike.
4. What are the “ports” on the lattice structure, right and left? I suggest adding few words in explanation of the meaning. For example, page 5, line 156 “will transmit to the right port” to be changed to “will transmit to the right, where the light leaves the nanocavity through the output port”?
I suggest the following improvement of the figures:
Figure 1
1. On the figure 1a, the lattice should be composed of the triangular holes with different sizes, to represent the subject of the paper exactly.
2. The red dashed line on Figure 1a that depicts the hexagon is not about the subject of the paper, because the unit cell of the lattice with δ ≠0 has no hexagonal symmetry.
3. On figure 1b, what is the yellow area? What is the rose area? What is the blue area? What do the “Gap” titles mean?
Figure 2
1. The figure 2 should have first a scheme of two types of VES interfaces, type 1, and type 2.
2. What is polarisation of light on the figure 2c?
Figure 3, both caption and the text. See also the comments in the pdf file:
1. What is bonding and anti-bonding coupled VCSs? Please explain in text (page 5, line 176)
2. The scheme showing the main elements of the structure should be added to the figure, i.e., The main parts composing the structure are to be outlined, i.e. VES of type 1, VES of type 2, other interfaces, the size of the structure, the nanocavity, the input and output ports.
3. What is the abscissa axis on the figure 3, a? The “Solution number” makes no sense.
4. What is the inset ono the figure 1a?
5. The figure 3 f should be a figure 3 d, and move to the left accordingly, because it provides a markup for two other figures. The figures 3d and 3 e should be changed accordingly.
6. Figure 3 b and c – the arrows representing the direction of light propagation?
I hope it helps

Comments on the Quality of English Language
The language of the paper could be improved by using the consistent terms through the whole paper.
Round 3
Reviewer 3 Report
Comments and Suggestions for Authors
Dear authors,
Thank you very much for the good work done for a review.
Author Response

(The authors gave the same response as above.)
